

# Risk and protective factors for drug dependence in two Moroccan high-risk male populations

Anis Sfendla[1], Dina Lemrani[1], Britt Hedman Ahlström[2], Meftaha Senhaji[1] and Nóra Kerekes[2]

[1] Department of Biology, Faculty of Sciences, Abdelmalek Essaâdi University, Tetouan, Morocco
[2] Department of Health Sciences, University West, Trollhättan, Sweden

## ABSTRACT

**Background:** Substance use is linked to biological, environmental, and social factors. This study provides insights on protective and risk factors for drug dependence in two Moroccan, high-risk, male samples.

**Methods:** Data from the "Mental and Somatic Health without borders" (MeSHe) survey were utilized in the present study. The MeSHe survey assesses somatic and mental health parameters by self-report from prison inmates ($n = 177$) and outpatients from an addiction institution ($n = 54$). The "Drug dependence" and the "No drug dependence" groups were identified based on the Arabic version of the Drug Use Disorder Identification Test's (DUDIT) validated cutoff for identifying individuals with drug dependence, specifically in Morocco.

**Results:** The majority of participants who had at least high school competence (67.6%), were living in a partnership (53.7%), were a parent (43.1%), and/or had a job (86.8%) belonged to the "No drug dependence" group, while the presence of mental health problems was typical among the "Drug dependence" group (47.4%). A multivariable regression model ($\chi^2$ (d$f$ = 5, $N$ = 156) = 63.90, $p < 0.001$) revealed that the presence of depression diagnosis remains a significant risk factor, while a higher level of education, having a child, and being employed are protective factors from drug dependence.

**Discussion:** Findings support the importance of increasing academic competence and treating depression as prevention from the persistence of drug addiction in male high-risk populations.

## INTRODUCTION

The problem of substance use has become a worldwide burden, affecting not only individuals' health and well being but also society's economic development and environmental sustainability (*Cartwright, 2008*; *Chen & Lin, 2009*). In 2014, it was estimated that global drug use prevalence was 5.2%, reflected by 247 million people who had used drugs during the previous year, and while almost 29 million people had substance use disorder (SUD), only one in six received treatment (*United Nations Office on Drugs and Crime (UNODC), 2016*).

Corresponding author
Anis Sfendla,
anis.sfendla@gmail.com

An increasing trend in drug use has been detected in developing countries, with cannabis as the primary drug of abuse followed by amphetamine-type stimulants, ecstasy, and opium derivatives (*Odejide, 2006*; *Pletzer et al., 2011*). Generally, the amount of health lost related to drug and alcohol use is higher in low- and middle-income countries than those in high-income countries (*Collins et al., 2011*). For example, in 2009 SUD in low-income countries accounted for a loss of 6.5 disability-adjusted life years (DALYs), compared with the loss of 1.9 DALYs in high-income countries (*Collins et al., 2011*). While there are no newer reports presenting actual information specifically for developing countries, globally it has been shown that DALYs caused by psychiatric disorders including SUD increased from 7.8 in 2005 to 11 in 2013 (*Murray et al., 2015*), suggesting that the 6.5 DALYs reported specifically for developing countries for nine years ago, today is even more serious.

Morocco is a unique African Arabic developing country under robust European influences, with a dominant Islamic culture, which itself strongly regulates and punishes drug and alcohol use. However, Morocco is considered one of the main producers of cannabis resin worldwide (*International Narcotics Control Board (INCB), 2005*), which makes the concern over drug use and its previously suggested health and societal effects (*Kessler et al., 1994*; *Chen & Lin, 2009*) highly relevant. The latest prevalence of SUD reported in this country was 5.8% (*Kadri et al., 2010*), a comparable prevalence with the global number reflected by 5% in 2010 (*United Nations Office on Drugs and Crime (UNODC), 2016*).

Substance use is linked to multiple negative consequences, including co-occurrence of other psychiatric disorders, somatic health problems, social marginalization, and even criminality (*Kessler et al., 1994*). A considerable amount of research has affirmed that the negative outcomes of many psychiatric disorders (such as anxiety, depression, psychosis, and schizophrenia) are related to co-existing substance use, while abstainers may have more favorable outcomes (such as better health status, life satisfaction, and education achievements) (*Brook et al., 2002*, *2011*; *Lennings, Copeland & Howard, 2003*; *Ellickson, Martino & Collins, 2004*; *Fazel et al., 2009*; *Van Ryzin, Fosco & Dishion, 2012*). It is also well documented that the prevalence of substance use is high in samples of prison inmates (*Fazel, Bains & Doll, 2006*), and those living in bad social circumstances (*Spooner & Hetherington, 2004*).

Regarding interpersonal risk and protective factors, longitudinal studies have found that anxiety and depression are linked to the subsequent development of substance use, with nearly one-third of major depressive patients having a SUD (*Davis et al., 2008*). Moreover, it was found that being single and the absence of employment were the strongest societal risk factors of SUD (*Lennings, Copeland & Howard, 2003*; *Davis et al., 2008*; *Brook et al., 2011*; *Vivek et al., 2011*; *Stone et al., 2012*).

It is known that lower socioeconomic status of developing countries is associated with often insufficient detection (both in quantity and quality) of somatic and psychiatric problems and with the lack of therapy and rehabilitation possibilities (*Patel, 2007*). From a research standpoint, epidemiological studies on psychiatric disorders are quite rare in developing countries (*Dewing et al., 2006*; *Plüddemann, Myers & Parry, 2008*). It is even more difficult to find scientific works focusing on risk and protective factors for the
development of addiction in Moroccan samples. To our knowledge, only one study has investigated the associated factors of psychoactive substance use in Morocco. This study collected data from middle- and high-school students and revealed that male gender, secondary school level (13–16 years of age), smoking tobacco, living with family members who use tobacco, and feeling insecure within family environment were the risk factors predicting drug use (*Zarrouq et al., 2016*).

The scarcity of research indirectly limits the development of effective preventive and adequate rehabilitation and treatment programs. With the present work we aim to contribute to this research area, investigating protective and risk factors for drug dependence in two Moroccan samples with high risk to develop drug dependence: male prison inmates and male psychiatric outpatients with a previous SUD diagnosis.

## PARTICIPANTS AND METHODS

### Study design

The "Mental and Somatic Health without borders" (MeSHe) international project (http://meshe.se) aims to identify culture specific personality, behavioral, physical, and psychiatric symptoms to improve upon possibilities for an early identification of those who may require (social and/or psychological) help with their substance use. The project assesses data from different study populations with the help of a standardized survey (the MeSHe survey), which includes, besides other validated inventories, the self-reported Drug Use Disorder Identification Test (DUDIT) (*Berman et al., 2005*). Within the frame of this international project, data from two high-risk samples were collected in Morocco, during the period between June 2013 and July 2014. General Directors of the Prison Administration and Rehabilitation and the Hasnouna Association Drug Users Support center consented to the study.

### Participants

There were too few female participants in the clinical sample and no data collected from female inmates; therefore, data from male outpatients with SUD diagnosis ($n = 54$) and male prison inmates ($n = 177$) were utilized. Inmates were recruited from local "Toulal 2" prison in Meknes, Morocco; while outpatients were recruited from a medical and psychological prevention center in Tangier, Morocco. The prison inmate sample reflects 7.2% of those incarcerated during assessment; whereas, all of the male outpatients chose to participate (100%). For further clarifications about clinical and inmate sample, survey administration and study design please refer to *Sfendla et al. (2017)*.

### Measures

The Arabic version of DUDIT was recently validated (*Sfendla et al., 2017*). The DUDIT consists of 11 simple questions, which assess the frequency of drug use during the past 12 months via self-reports. For the identification of those with drug dependence (the "Drug dependence" group), the previously validated cutoff score of 3.0 was used (*Sfendla et al., 2017*); therefore, those scoring in the DUDIT zero, one, or two points formed the group named as the "No drug dependence."

The MeSHe survey includes socio-demographic variables (education, employment, partnership, and parenthood) in its background questionnaire section. Education levels were categorized into groups of "high education" (that is, those who achieved high school and/or college/university education) and "low education" (participants reporting no education or only elementary/secondary school education). Employment was recorded based on a binary reply where current job was indicated (Yes/No). For inmates, the answer should have reflected the state prior to incarceration. Partnership status was categorized into "living in a partnership" if the participant indicated that he was married or living with a partner; and "single" when the participant indicated that he was divorced, separated, or living alone. Finally, parenthood was also recorded into a binary variable based on having or not having children.

The MeSHe survey also assesses information about the existence of previously received (from a physician) psychiatric diagnosis of any of the following: depression, anxiety disorder, obsessive-compulsive disorder (OCD), post-traumatic stress disorder (PTSD), bipolar disorder, other personality disorders, eating disorder, or schizophrenia.

## Statistical analysis

When comparing descriptors between the groups of "Drug dependence" and "No drug dependence," Student $t$-test was used for continuous variables (age and drop-out education age), while the chi-square test ($\chi^2$) was used for categorical variables (education, profession, marital status, children, and psychiatric disorders), where the degree of the association was assessed by the *phi* coefficient ($\Phi$). Cohen's criteria were applied to *phi* coefficient where 0.10 indicated a small effect, 0.30 a medium effect, and 0.50 a large effect (*Cohen, 1988*). When the chi-square assumption was violated, likelihood ratios were used. Risk Ratios (RR) were reported reflecting the probability of different psychiatric disorders in the "Drug dependence" group divided by the probability of psychiatric disorders in the "No drug dependence" group. Only significant descriptors were placed in the multivariable prediction model. The multivariable logistic regression model was used to measure their association to drug dependence. The dependent variable was coded 0 for those belonging to the "No drug dependence" group and 1 for those belonging to the "Drug dependence" group. Thereby, a negative beta coefficient would indicate that an increase in the continuous variable or a "yes" answer for the categorical variables are associated with a decrease in the probability of a score of 1 in the independent variable (being drug dependent). Consequently, a positive beta coefficient suggests an increased probability of being drug dependent with increased continuous variable or with the presence (yes) of any categorical variable. All statistical analyses were conducted by SPSS version 21 for Windows, with significance level at 5%.

## Ethical considerations

The study process was in agreement with the Helsinki declaration (*World Medical Association, 2008*). Participation was voluntary and anonymous. The cover page of the survey informed the participants about the questionnaire's content and the aims of the study and they also received oral information, after which they could decide whether

they want to answer the survey. All participants were assured that their answers would not affect sentences (in case of inmates) or their treatment plans (for those who were outpatients). Completion of the survey was considered as consent of participation. The use of the survey was approved by the Directorate-General of Prison Administration and Rehabilitation (for inmates) (case identifier number: 13993), and by the Director of the Medical and Psychological Prevention Center (for those who were outpatients).

## RESULTS

The clinical outpatients sample mean age was 38.3 (SD = 8.3), and the mean age of the prison inmate sample was 30.8 (SD = 10.6). No significant differences were detected in age ($p = 0.73$), the age when participants finished education ($p = 0.65$), and the prevalence of selected psychiatric problems between these two samples ($p = 0.25$).

### Comparison between "Drug dependence" and "No drug dependence" groups

Significant differences were found when separating participant with drug dependence from those without drug dependence (Table 1). Fewer participants in the "Drug dependence" group lived in a partnership, had a child, had a high level of education, or were employed, compared to the "No drug dependence" group ($p < 0.001$, in each case). In detail, over two-thirds (67%) of those indicating active drug use and drug dependence during the past 12 months had a low education level, compared to only one-third (32%) of those who reported no or minimal (less than three points in DUDIT) drug use. The chi-square test for independence indicated a significant difference, with a medium effect size, in the level of education ($\chi^2$ (1, $n = 189$) = 21.3, $p < 0.001$, $phi = 0.34$) between the two groups. Most (82%) participants from the "Drug dependence" group and more than half (57%) from the "No drug dependence" group reported being single, and in a similar ratio they reported not having children (80% and 46%, respectively). Chi-square tests for independence indicated a weak but significant difference in marital status ($\chi^2$ (1, $n = 190$) = 13.2, $p < 0.001$, $phi = 0.26$) and in parenthood status ($\chi^2$ (1, $n = 184$) = 21.4, $p < 0.001$, $phi = 0.34$) between those with drug dependence and those who were not active or no users. Similarly, the unemployment rate was more than three times higher in the "Drug dependence" group (43%) than in the "No drug dependence" group (13%). The chi-square test showed a significant and medium-strong association between active drug dependence and unemployment ($\chi^2$ (1, $n = 187$) = 17.4, $p < 0.001$, $phi = 0.30$). While the presence of any psychiatric disorder was not significantly different between the groups (47% and 38%, respectively), a comparison between the prevalence of defined disorders revealed that depression was a significantly more frequent coexisting problem in those with drug dependence (22%) compared to those with no dependence or no use (3%) ($\chi^2$ (1, $n = 167$) = 9.96, ($p < 0.001$), $phi = 0.24$).

Admittedly, RR revealed that the "Drug dependence" group had 6.4 times (95% CI [5.351–7.654]) higher risk of depression compared to the "No drug dependence" group. Likewise, the risk of having OCD was 2.7 times (95% CI [2.699–3.434]) higher, and the risk of having schizophrenia and personality disorder doubled, while the risk of

**Table 1 Socio-demographic and clinical background of participants.**

| Background variables | Total sample[h] % (n) | Drug dependence group[f] (n = 119) | No drug dependence group[f] (n = 72) | RR[g] | t/χ² | p-value | Φ |
|---|---|---|---|---|---|---|---|
| Age M (sd) | 32.7 (10.62) | 32.52 (10.95) | 34.01 (10.08) | | 0.935 | 0.73 | |
| Education dropout age | 16.95 (7.7) | 15.93 (4.29) | 19.45 (3.61) | | 4.21 | 0.65 | |
| Education[a] % (n) | | | | | | | |
|   High educational level | 46% (87) | 33.1% (39) | 67.6% (48) | | 21.31 | <0.001 | 0.34 |
|   Low educational level | 54% (102) | 66.9% (79) | 32.4% (23) | | | | |
| Marital status[b] | | | | | | | |
|   In a relationship | 27.9% (53) | 18.6% (22) | 43.1% (41) | | 13.25 | <0.001 | 0.26 |
|   Single | 72.1% (137) | 81.8% (96) | 56.9% (31) | | | | |
| Parenthood | | | | | | | |
|   Yes | 32.6% (60) | 20.5% (24) | 53.7% (36) | | 21.4 | <0.001 | 0.34 |
|   No | 67.4% (124) | 79.5% (93) | 46.3% (31) | | | | |
| Employment | | | | | | | |
|   Yes | 67.9% (127) | 57.1% (68) | 86.8% (59) | | 17.42 | <0.001 | 0.30 |
|   No | 32.1% (60) | 42.9% (51) | 13.2% (9) | | | | |
| Psychiatric disorders[c] | | | | | | | |
|   Yes[c] | 44.1% (79) | 47.4% (54) | 38.5% (25) | | 1.33 | 0.25 | |
|   No[c] | 55.9% (100) | 52.6% (60) | 61.5% (40) | | | | |
| Depression | 15% (25) | 21.5% (23) | 3.3% (2) | 6.44 | 9.96 | 0.002 | 0.24 |
| Anxiety | 20.9% (37) | 21.2% (24) | 20.3% (13) | 1.04 | 0.02 | 0.88 | |
| OCD[d] | 10.6% (18) | 13.6% (15) | 5% (3) | 2.72 | 3.06 | 0.08 | |
| PTSD[e] | 19% (34) | 19.1% (21) | 18.8% (13) | 1.01 | 0.00 | 0.97 | |
| Bipolar disorder | 3.4% (6) | 3.8% (4) | 3.3% (2) | 1.17 | 0.036[Lh] | 0.85 | |
| Eating disorder | 21.1% (38) | 18.8% (21) | 25% (17) | 0.75 | 0.992 | 0.32 | |
| Schizophrenia | 5.7% (10) | 7.3% (8) | 3.1% (2) | 2.36 | 1.46[Lh] | 0.23 | |
| Personality disorder | 9.4% (16) | 11.1% (12) | 6.3% (4) | 1.75 | 1.06 | 0.30 | |

Notes:
[a] Education was divided into *high educational level*, including those with high school and /or university/college education, and *low educational level*, including those with no education or only elementary/secondary school education.
[b] Living in partnership includes the situations of being married, remarried, and living with a partner. Not living in a partnership comprises being single, divorced, and separated.
[c] The existence of Substance Use Disorder diagnosis (SUD) was excluded from psychiatric diagnoses.
[d] Obsessive Compulsive Disorder.
[e] Post-Traumatic Stress Disorder.
[f] Drug dependence was classified based on DUDIT total score using cutoff score (Cutoff ≥ 3 drug dependence) and (Cutoff < 3 no drug dependence).
[g] Risk Ratios reflecting the probability of different psychiatric disorders in the Dependent group divided by the probability of psychiatric disorders in the Non-dependent group.
[h] Listwise deletion was used to handle missing values resulting in different total sample for each analysis and consequently different percentages.
[Lh] Likelihood ratio test was used based on the violation of chi-square assumption.

having a coexisting eating disorder was 25% lower in the "Drug dependence" group compared to the "No drug dependence" group.

## Multivariable prediction model

Only those variables that significantly differed between the groups were fitted in the multivariable prediction model. The final model included the following factors: education level, having any children, being employed, being in a partnership, and existing diagnosis of depression. Table 2 presents the multivariable associations of sociodemographic and

**Table 2 Multivariable model for prediction of drug dependence.**

| Predictors | β | p | $e^B$/OR | 95% CI |
|---|---|---|---|---|
| Education | −1.862 | <0.001 | 0.15 | 0.065–0.374 |
| Marital Status | −0.531 | 0.41 | 0.59 | 0.167–2.075 |
| Parenthood | −1.295 | 0.039 | 0.27 | 0.080–0.937 |
| Employment | −1.709 | 0.001 | 0.18 | 0.065–0.505 |
| Depression | 2.821 | 0.001 | 16.79 | 3.001–93.968 |

Notes:
Table of the multivariable logistic regression model that was used to measure association to drug dependence.
Model Summary; ($\chi^2$ (5, $N = 156$) = 63.90, $p < 0.001$).
CI, Confidence interval; OR, Odds Ratio.

clinical correlates with active drug dependence. The model containing all predictors was statistically significant ($\chi^2$ (d$f = 5$, $N = 156$) = 63.90, $p < 0.001$), showing that the model could explain over 60% of drug dependence. Four of the predictor variables (education level, existence of employment, having any children, and diagnosis of depression) had a highly significant contribution to the model. The strongest predictor of drug dependence in this model was the existence of a diagnosis of depression. Participants who had reported the existence of a diagnosis of depression were 17 times more likely to be also identified by the DUDIT as people who have active drug dependence. On the other hand, high education level, having a child, and being employed were significant protective factors from being active in drug use and dependence. Participants with a higher education profile were about seven times less likely to have a drug dependence. Having children and being employed also decreased the likelihood of being a drug dependent person by about four times and five-and-a-half times, respectively.

## DISCUSSION

This study contributes to the rarely studied field of drug addiction in Morocco and provides insights into protective and risk factors utilizing data from Moroccan samples of prison inmates and individuals with a previous diagnosis of substance use disorder (SUD), both samples with a high risk of active drug use and dependence. In these Moroccan high-risk populations, rsquo; education level, employment, partnership, parenthood, and the diagnosis of depression were significantly associated with being actively drug dependent. Specifically, the study showed depression is the factor that has the strongest association with drug dependence, while higher level of education, being employed, and having a child can be significant protective factors against it.

Considering socio-demographic factors, previous studies have also indicated that high level of education is a protective factor from substance use (*Fitzpatrick, Piko & Wright, 2005*; *Mirza & Mirza, 2008*; *Piko & Kovács, 2010*). While these studies assessed protective factors among younger samples and in other nations, results in the present study confirm the same findings in high-risk Moroccan populations, specifically indicating that education is the strongest protective factor even in the samples of clinical and incarcerated individuals. Another significant protective factor in high-risk samples was employment. Previous empirical evidence has persistently reported that unemployment is often coupled

with substance use (*Henkel, 2011*; *Compton et al., 2014*; *Lee et al., 2015*). In addition, international research has previously shown a negative association between drug use and employment (*French, Roebuck & Alexandre, 2001*; *DeSimone, 2002*). The association between unemployment and drug use has been suggested to be mediated by personality traits—such as stress reactivity or impulsivity—that could relate unemployment to substance use (*Compton et al., 2014*). Assuming that the prison inmate sample may consist of individuals with a high level of impulsivity (*Cuomo et al., 2008*; *Bernstein et al., 2015*), then the previously suggested interaction could also be tested in this study; however no direct information was available about the level of impulsivity or impulse control in the Moroccan samples.

The results support previous statistics, showing that the proportion of adults reporting active use of illegal drugs is increased with unemployment (*Badel & Greaney, 2013*).

Of the other socio-demographic factors, while relationship was not a significantly affecting factor of active drug dependence in the Moroccan high-risk populations, parenthood emerged as an important protective factor. This is in accordance with previous findings showing that parenthood is most often linked to limited or no drug use, and therefore custodial parenthood results in the decline of the risk of drug use (*Bachman et al., 2002*; *Fergusson, Boden & John Horwood, 2012*). It is important to mention that age by itself was not correlated to substance use; therefore, the protective action of parenthood and employment is not mediated by older age.

As regard to the existence of (any) psychiatric problem, no association was found with drug dependence. However, the presence of depression significantly increased the risk of coexisting drug dependence in high-risk samples. This result is in accordance with previous findings showing that drug use and depression coexist in clinical and prison inmate samples (*Ross, 1988*; *Rowe et al., 1995*; *Vreugdenhil et al., 2003*; *Najt, Fusar-Poli & Brambilla, 2011*). While many studies have confirmed the association between depression and drug use, the explanation of their link is still not complete. *Bovasso (2001)* found that adult participants with an initial diagnosis of cannabis use are four times more likely to have depressive symptoms at a follow-up assessment than those with no diagnosis of cannabis use. Other longitudinal studies have also suggested that substance use precedes depressive symptoms even in adults (*Weller & Halikas, 1985*; *Grant, 1995*; *Angst, 1996*).

The findings of the present study are consistent with the results of previous studies on risk and protective factors, and also of high relevance to substance abuse treatment and prevention programs targeting prison inmates and psychiatric patients. Outpatient centers should help individuals to find social support and job opportunities, improve their educational level, and provide adequate medical care even for their depressive symptoms, which is possible when utilizing integrative treatments (*Weisner et al., 2001*). Also, our work by showing that social factors are indeed protective factors for drug dependence even in high risk groups, supports the newly launched theories questioning the dominant influence of the biological, "brain disease model of addiction", which try to suppress the psychosocial perspectives of addiction (*Heather et al., 2018*).

While the present study is highly important due to the scarcity of psychiatric data collection and studies in developing countries, it also has several limitations. The DUDIT

only refers to the pattern of use and did not make possible specific identification of the type of drug use. Not knowing which substance or substances the participants in this study were using may result in confounding with respect to which psychiatric conditions were of most concern. This should be considered in the general recommendations for community treatment services. The relatively small sample size within each subpopulation could be argued to cause possible type II errors in our analyses, and definitely limit the generalizability of the results, mainly considering prison inmates as only 7.2% of the total incarcerated population participated in the study. Importantly, the study sample included only male participants, therefore the findings and recommendations may not be relevant to the female population. Moreover, the cross-sectional design of the MeSHe study limits the possibility of causality analyses. Finally, the existence of psychiatric diagnoses was self-reported and not gathered from inpatient registers, but the study design (anonymous participation) does not allow any follow-up or merging of the collected data.

Based on these limitations it could be recommended that a larger study be designed, which addresses the potential selection bias raised above, and which incorporates measures that will make separate analyses possible for different types of drugs. Further research investigating the effectiveness of treatment for co-occurring substance use and psychiatric disorders in a low- and middle-income country setting, such as Morocco, is also warranted.

## CONCLUSION

The results of the present study emphasize the need for adequate psycho-educational programs alongside the established medical treatments and social support for individuals with drug dependence. Our study shows that integrated treatment is required for addiction, where not only one medical problem is addressed (in this case SUD), but also comorbid psychiatric problems (such as depression); importantly, these should always be completed with education and social-psychiatric care. In Morocco, medical and psychological prevention centers are linked to associative work mainly under the frame of risk reduction, psychosocial support, and employment integration for drug users. There is a need to increase the number of these outpatient care centers with qualified personal in psychiatry, also in Morocco, to be able to decrease the prevalence of drug use and dependence, consequently reducing the prevalence of negative results (such as criminality, chronic mental illness, self-harm behavior, and even early death).

### Funding
The authors received no funding for this work.

### Competing Interests
The authors declare that they have no competing interests.

## Author Contributions

- Anis Sfendla has contributed with data collection in the prison population, with data analyses as well as responsibility for writing and revising the manuscript.
- Dina Lemrani has contributed with data collection in the inpatient, clinical population and with revising the manuscript.
- Britt Hedman Ahlström has contributed in writing and revising the manuscript.
- Meftaha Senhaji is the senior supervisor of the data collection in Morocco. She has coordinated the studies in the prison and clinical populations. Further she has contributed to the study with critically important intellectual feedback on interpretation of our results as well as on revision of the manuscript.
- Nóra Kerekes designed the "MeSHe" project from which data has been used for the present study. She has contributed in design of the study, interpretation of data as well as to continuous monitoring of progress and revision of the manuscript.

## Human Ethics

The following information was supplied relating to ethical approvals (i.e., approving body and any reference numbers):

General Directors of the Prison Administration and Rehabilitation and the Hasnouna Association Drug Users Support center consented to the study (case identifier number: 13993).

## Data Availability

The raw data are provided in a Supplemental File.

## Supplemental Information

Supplemental information for this article can be found online at http://dx.doi.org/10.7717/peerj.5930#supplemental-information.

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
