# Peer review of "Risk and protective factors for drug dependence in two Moroccan high-risk male populations"

_PeerJ, doi:10.7717/peerj.5930_

## Round 0.1 · original submission · Major Revisions

You will see that the reviewers have recommended a number of major revisions which will help strengthen the paper. Please pay particular attention to language - both in the area of the terms used for your study participants and those used for drug use. Both reviewers commented on this.

Please also fully acknowledge the limitations regarding use of the DUDIT highlighted by Reviewer 2, as well as the recommendations for future research.

Reviewer 1 ·

Basic reporting

INTRODUCTION
This section was well written and provided a basic rationale for conducting the study. I thought, however, that the introduction did not adequately justify why incarcerated people specifically were included – and this could possibly be brought to the forefront more in future iterations. Also, did the participants provide written, informed consent prior to taking part in the study? If so, this should be mentioned. If not, it should be justified.

• Line 65: Do the authors have a newer reference to support this (seemingly time-relevant) claim?
• Line 67: When discussing a loss of 6.5 DALYs, presumably this is per person? If so, this should be explicitly mentioned.
• Lines 71-72: Is a prevalence of 5.8% really very different to one of 5.0%? I would question the strength of this claim.
• Lines 86-88: Do the authors have a referent to support this assertion?

Experimental design

MATERIALS AND METHODS
- Adequate design and clear research questions.

• Line 131: It was interesting to read that those participants who finished high school were considered to be in the “high education” group; is this higher than the population demographics of Meknes would suggest?
• Line 137: What about participants whom had girlfriends not living with them? Were they counted as single, despite being in a relationship?
• Line 152: The name “Non active substance abuse” in this sentence appears to be different from the normal name used in the rest of the manuscript.

Validity of the findings

RESULTS
• Lines 182-183: It would be helpful if the authors could briefly indicate the meaning of this finding (i.e., the direction of the association); although it is implied, it is not overly stated.
• Lines 186-188: Similarly here, the direction of the association would be helpful to avoid ambiguity.
• Line 198: The authors should report the 95% Cis when reporting the RRs.
• Line 214: Should ‘screened for active substance use’ read ‘screened positive for substance use’?
• Line 217: And the same here, also?

DISCUSSION
This was also well written and it was good to see the authors transparently discussing the potential limitations of their research design and resulting manuscript. Clinical implications and directions for future reswarch

• Line 235: The phrase “A large margin of empirical evidence” does not make sense – presumably the authors are referring to the large body of literature or large amount of evidence?
• Line 242: Can the authors justify with references their assertion that incarcerated males have high levels of impulsivity? This is almost certainly a true statement, but backing it up with some evidence would be preferable.
• Line 252: The name “John” is not needed here and should be removed. Line 272: “DUDIT only refer” should be “The DUDIT only refers”.
• Line 281: “need for”, not “need of”.

Additional comments

Thank you for the opportunity to review this manuscript, which examined the risk and protective factors for substance use in a sample of incarcerated males and a sample of outpatients from a substance use facility in Meknes, Morocco. These are two Moroccan samples that are relatively under-represented in the literature and so it is encouraging to see research studies being conducted in this space. This manuscript was mostly well-written and I imagine that the topic would likely be of interest to PeerJ’s readership. I do have a few specific points for the authors to consider (see above) which I believe may strengthen the manuscript:

TERMINOLOGY:
• Some of the terminology used in this manuscript (such as ‘addicts’, ‘inmates’, and ‘not active or no abuser’) are all potentially quite reductionist and stigmatising in nature, reducing the person to his offending history or substance use only. I would strongly encourage the authors to remove references to addicts and inmates, and perhaps amend ‘no abuser’ to ‘no abuse’ when describing their categories.
• Unless it is a PeerJ requirement, I would recommend removing the word ‘subjects’ and replacing it with ‘participants’ throughout the manuscript, as subjects is quite outdated.
• Data are plural – so it should be “data are”, “data were” etc. instead of “data was collected”.

ABSTRACT
• It would be helpful to the reader to understand (even briefly) what ‘academic competence’ refers to and how it was measured.
• In the Results section of the abstract, having some figures and statistics would be helpful to immediately orient the reader to the strength of the observed associations.

I hope this feedback is of use and I wish the authors the best of luck in their future endeavours.

Reviewer 2 ·

Basic reporting

This manuscript describes a study investigating risk and protective factors of drug dependence in two groups of high risk Moroccan males using cross-sectional self-report survey data. The authors highlight the paucity of evidence in this area for low/middle income countries (LMIC), and especially in Morocco, whilst acknowledging that there is a broad range of literature on this topic investigating these issues in other settings. Whilst there is some literature based on research in other LMIC that may be relevant, the rationale for this paper is sound overall. There are some limitations both with the study design, statistical analyses and the measures used that are largely acknowledged in the discussion that limit the scope of the research questions and findings. There is also some area for improvement with respect to sentence structure, language and terminology that would improve the quality of the manuscript. These issues are considered below, and some specific suggestions are offered to assist the authors.

The language used needs to be reconsidered. Terms such as “drug abuse” and “addicts” and “suffered” are now considered to further stigmatise people who are experiencing problematic drug use or dependence. Using terms such as “people experiencing drug dependence”, or “problematic use” or other such non-stigmatising language should be considered.

The terms “drug use”, “drug abuse” and “drug dependence” appear to be used interchangeably throughout the manuscript. It is unclear whether these are referring to the same thing, or different categories of drug use. The authors should considering clarifying this, and if they refer to the same thing, should use the same term throughout the manuscript.

The grammar and language can be difficult to follow in places. For example, the sentence beginning on line 70 and ending on line 72, as well as the sentence on lines 86-88.

There are several sentences in the manuscript that make assertions or statements of fact that do not include a reference. Often these are followed by a sentence that does, however every statement of fact or assertion should have a reference to support it. For examples, see lines 57-59, 65-66 and 86-90 of the manuscript.

There is literature referred to as evidence of drug use trends that are more than 10 years old (eg, Odejide 2006 on line 65). There is more recent research published that would be more appropriate and accurate to refer to. Odejide et al also solely report on drug use in Africa, so any research investigating other developing countries would help to strengthen this section.

The burden of disease literature cited in the manuscript in Collins 2011 is probably getting a bit dated now, and there has been more recent work published in this area, including “Estimating the true global burden of mental illness” by Vigo et al (2016) published in Lancet Psychiatry. It would strengthen the manuscript if more recent literature was referred to.

The authors refer to several studies into protective and risk factors for drug dependence, and also the association between psychiatric disorders and drug dependence, then suggest that there is a scarcity of epidemiological research on psychiatric and substance use lower-middle income countries (LMIC) as a key part of their rationale for their study, suggesting that this is preventing the development of treatment programs. This appears to be contradicted by other research from LMIC that assess treatment programs, including: Bosman et al 2014, http://www.psychology.uct.ac.za/sites/default/files/image_tool/images/117/Caroline.Bosman.pdf; Gouse et al 2016, doi:10.1371/ journal. pone.0147900; Magidson et al 2017, doi:10.1080/08897077.2017.1380743

There is a reasonable amount of epidemiological research published in this area that comes out of South Africa that may be relevant to the research questions being investigated in the current study. Some of this work investigated substance use (see eg Pluddemann et al 2008, DOI:10.1080/09595230701829363; Parry et al 2015, https://doi.org/10.15288/jsa.2002.63.430; Dewing 2006, https://doi.org/10.1080/09687630500480228; Myers et al 2010, https://doi.org/10.1186/1747-597X-5-28,

Other research has investigated various aspects of mental illness and effective service provision in LMIC (see eg van Ginneken et al (2011) doi:10.1002/14651858.CD009149; Chibanda et al (2014), doi: 10.1097/QAI.0000000000000258). Some of this research seems relevant to the current study and may be worth mentioning to provide context.

The survey provided is in Arabic, so I am unable to understand what questions were asked, it would be helpful if an English version was also provided.

Experimental design

The research question is well expressed, investigating protective and risk factors of drug dependence in two “high risk” groups. However, the authors may consider defining what a “high risk” group is, as this is not clear. Risk has a technical meaning in epidemiology, so defining this is essential, as the obvious question is high risk of what?

The authors state that there is particular dearth of research looking at addiction in “Moroccan samples”. It is unclear why research done on non-Moroccan samples like some of the LMIC examples above would not be relevant, and may potentially weaken the rationale for doing this project. Articulating what gaps the current project will address and how it differs from previous LMIC research a little more clearly may strengthen the rationale for this study. Some background on what makes Morocco unique from other LMICs (other than the high production rates of cannabis resin) may be of value.

It is unclear why the high rates of cannabis resin production in Morocco “makes the concern over the health and societal effects of drug use highly relevant”. Is this because there are high rates of cannabis use in Morocco? If so, what are those rates? Also, is there evidence of substance use-related poor health outcomes in Morocco? If so, what does that evidence tell us about these poor health outcomes?

The authors should consider replacing the heading “subjects” with “participants”, this is consistent with modern convention.

The analyses conducted seem mostly appropriate, however there are a couple of things that the authors could consider to strengthen them. The first is the method of running univariate comparisons (via chi square tests in this case) and then solely relying on these results to build a multi-variate model is becoming outdated and does not always control sufficiently for potential confounders. The univariate analyses are a great starting point, but there is also a large amount of literature (some of which the authors refer to) about known associations between certain psychiatric disorders and substance use. Combining what is already known from the literature, with univariate analyses when building the multivariate logistic regression model is one way of ensuring that the model is as robust as possible. The second thing that could be done is to use a directed acyclic graph (DAG) to assist with this process (Shrier 2008, https://bmcmedresmethodol.biomedcentral.com/track/pdf/10.1186/1471-2288-8-70).

Validity of the findings

The limitation of the DUDIT to a binary yes/no for drug dependence, without being able to identify the specific drug is a potentially fatal one for this paper. Clearly different drugs react differently and are often associated with different psychiatric conditions (ie methamphetamine with psychosis and anxiety and possibly depression, cannabis with depression and possibly psychosis, alcohol with depression etc). Depression is likely the most common psychiatric problem associated with the use of a range of different drugs. In clinical practice, there are different treatments required depending on the type of drug that has been used problematically. Not knowing which substances the participants in this study were using may result in confounding with respect to which psychiatric conditions were of most concern. For example, a small number of participants may have used stimulants, and may report schizophrenia, while a larger number of people may have been drinking alcohol or using cannabis, and may report depression as their main problem. Without being able to analyse these groups separately, clearly the problem reported by the larger group is going to appear more prevalent. This has implications for making general recommendations for community treatment services. Services that treat alcohol or cannabis-related psychiatric problems may not be appropriate or effective for people with stimulant-related psychiatric problems. Simply treating depression for example may not be effective. There is also no way of measuring polysubstance use in the group, as the psychiatric issues for the use of multiple drugs concurrently compared to one drug may be quite different again. Acknowledging these limitations is important, and qualifying any recommendations made for changes to treatment services with these limitations in mind would improve the credibility of the findings.

There is possible selection bias in the sample that is recruited from the prison, as only 7.2% of the total incarcerated population participated, whereas all outpatients approached participated. It is well known that people who volunteer are often systematically different in profile to those who do not. This is a limitation of this study and could potentially result in confounding, something that should be acknowledged.

Another suggestion with respect to the model is based around the fact that the difference in OCD reported between the groups is approaching significance (p=0.08). The authors may consider conducting a sensitivity analysis by running the model with OCD included to see if it changes their results.

An observation of the findings is the possibility that anxiety may be under-ascertained in the participant group, as previous research suggests that this often co-occurs with depression. It is possible that the authors have already considered this and investigated whether participants reported these conditions together or whether only one or the other was reported in most cases. This would be interesting to know from a clinical perspective.

It appears that the finding that depression is the “strongest predictor of ongoing, active drug abuse” may need to be reconsidered. This implies that depression is causal and predictive of future drug use (which it is impossible to conclude using these cross-sectional data). It may be that participants reporting depression is associated with participants also reporting long-term drug use up until the date they participated in the survey. From these data it is impossible to conclude whether depression or drug use is the dependent variable so it is just an association, and nothing “ongoing” can be concluded without follow-up data.

Line 235 “a large margin of empirical evidence” is confusing, consider re-wording as it is unclear what a large margin refers to. In the same sentence “unemployment is coupled with” may also be better rephrased as “unemployment is associated with”.

Line 241, what evidence is there to assume that prison inmates may be impulsive? Referring to this evidence may clarify and improve this statement.

Line 245, it seems unclear how the results from the current study support a hypothesis that employment can move substance users from a “drug culture” to a “mainstream culture”. It is impossible to test this hypothesis with the current study design, a randomised trial would be required to do this. This appears to be speculation, which is permitted by the journal, but should be acknowledged as such. The authors could also reconsider the wording of this paragraph, particular the terms “drug culture” and “mainstream culture” as these implicitly carry value judgements as they are not easy to define.

The paragraph beginning on line 245 is a long one and could potentially be split up. It also could be cut down in size and put more succinctly so that the points being made can be done with less words.

Line 266, consider rewording this to something like “the findings of the present study are consistent with the results of previous studies…”

Line 268, it is not clear what the rationale for these recommendations is. There is no evidence presented to suggest that helping participants with social support and job opportunities will assist to treat substance use disorders and psychiatric illness. A participant reporting poor education and unemployment may be associated with a number of underlying causes or exposures from childhood up until the time of the study. One suggestion is to refer to the literature for evidence-based interventions that have been successful in assisting with treating co-occurring substance use and psychiatric disorders. Providing a reference for the evidence that adequate medical care that addresses depressive symptoms (or depression?) is possible when using integrative treatments would also strengthen this section.

Line 275-276, consider rewording for clarity. The research design means that only a male-specific discussion of the results is possible, so that the findings and recommendations may not be relevant to the female population.

Additional comments

The authors may consider recommending that in light of the small sample size and limited power of their study, that a larger study be designed that addresses the potential selection bias raised above, and that incorporates measures that will make separate analyses possible for different types of drugs. Measures such as the WHO ASSIST are useful to collect data of this type. Further research investigating the effectiveness of treatment for co-occurring substance use and psychiatric disorders in a LMIC setting such as Morocco is also likely warranted.

---

## Round 0.2 · Minor Revisions

Please amend the title to read as follows:

Risk and protective factors for active substance abuse in two Moroccan high-risk male populations

Please also attend to the comments of Reviewer 2 as these will help strengthen the paper.

Reviewer 1 ·

Basic reporting

No comment.

Experimental design

No comment.

Validity of the findings

No comment.

Additional comments

Thank you for the opportunity to review the revised version of this manuscript. I am satisfied with the changes that the authors have made on the basis of my suggestions and I wish them luck in their future endeavours.

Reviewer 2 ·

Basic reporting

Nothing to add here

Experimental design

Nothing to add here

Validity of the findings

Nothing to add here

Additional comments

Reply to author’s response to initial comments:

In my opinion, the manuscript has been improved by the many changes the authors made, and is much stronger and more balanced than the previous version.

There are a few of the authors’ comments that I would like to address as I think they are important issues that were not resolved by the authors’ response to my initial comments, and addressing these issues will strengthen the manuscript:

“Thank you. We have now rephrased these sentences and now they read as follows: “The latest prevalence of drug use reported in this country was 5.8% (Kadri et al., 2010), a somewhat higher prevalence when compared to global numbers reflected by 5% in 2010 (UNODC, 2016).” (Lines 70-73) & “It is known that lower socioeconomic status of developing countries is associated with insufficient detection of somatic and psychiatric problems both in quantity and quality ; and also with lack of therapy and rehabilitation possibilities (Patel, 2007)” (Lines 91-93).”

I suggest that 5.8% is similar rather than somewhat higher than 5.0%. You could refer to it as marginally higher, but it is almost the same, so saying somewhat higher may be misleading.

“We would though humbly disagree with the suggestion to use terms “people experiencing drug dependence”, or “problematic use” as drug dependence is a psychiatric disease not an experience”.
“Cancer patients or patients with diabetes can be called diabetics and cancer patients without worrying about stigmatization. Being dependent of drugs/substances is another type of disease and we believe that not the word “addicts” which should be changed but the stigmas and indictments coupled to this word.”

I respectfully disagree with these statements. Whilst there has been some previous support for using a model of addiction/dependence as a disease, the emerging expert opinion is that this idea has provable flaws and is now outdated. There is no consensus yet on exactly how to characterize addiction, however it appears that most experts are moving away from using a model of addiction as a disease and I would suggest that the authors do this as well. For some examples of recent literature that discusses this, please see the following for the different perspectives: DOI 10.1007/s12152-016-9293-4; DOI 10.1080/16066359.2017.1399659; DOI 10.1007/s12152-017-9303-1
Also, please refer to “Treatment modalities for pregnant women with opioid use disorder” by Gressler et al, published recently in the Lancet for an example of how people who essentially have a problem with addiction are described in the academic literature in leading journals. They describe “pregnant women with opioid use disorders” not “opioid addicts” or “pregnant opioid addicts”, this is now normative in the addiction research area and I would suggest that these conventions are followed and the manuscript is changed accordingly.
Similarly, the terms “diabetics” and “cancer patients” are not used in the leading health literature any longer, rather “people with diabetes” and “people/patients with cancer” are used as these are considered less reductionist terms. Please see http://dx.doi.org/10.1016/S0140-6736(18)30314-3 and https://doi.org/10.1016/S0140-6736(18)31078-X for two recent examples of this published in the Lancet.

“Very interesting suggestion. Two thoughts about it: the first is that as previously mentioned, if it is possible, we hope not to complete the present study with further statistical analyses. The second is that difference in the prevalence of Anxiety disorder could not be measured at all (p=0.88) and as OCD is a type of anxiety disorder there is high possibility that further analyses would not shed lights for other associations (from our data).”

OCD has its own section in the DSM-V, and while the preamble section provided in the manual suggests that some OCD disorders have a close relationship with anxiety disorders, it is clinically not correct to state that OCD is a type of anxiety disorder.
The suggestion to re-run the model with an extra covariate was made based on the assumption that this would be a simple adjustment to the syntax used to analyse the data originally, particularly for a small dataset like the one this study has used. This sensitivity analysis could be easily reported in 1 short sentence in the results if there is found to be no difference, and if there is found to be a difference, then this is an important outcome that will change the results and make the paper stronger, so in my opinion would justify any extra effort required. This seems like a very simple and reasonable suggestion to improve the robustness of the analyses and is encouraged unless there is some reason why this would be difficult or not possible to do.

I had some additional minor suggestions based on reviewing the revised manuscript:

Line 101-102 “feeling of insecure” could be changed to “feelings of insecurity” or “feeling insecure”
Line 107, should the word “samples” or “groups” be inserted after Moroccan?
Line 243, I would consider changing “ongoing, active drug abuse” to “active drug abuse”. The word “ongoing” implies that you have measured drug abuse at more than 1 time point, which is not the case. I would suggest just reporting “active” drug abuse because this is what you measure at the 1 point in time when the data were collected for each person.

---

## Round 0.3 · accepted · Accept

The reviewers are satisfied with the revisions and we are able to accept the paper.

# Reviewer 2 ·

Basic reporting

Nothing to add

Experimental design

Nothing to add

Validity of the findings

Nothing to add

Additional comments

The authors have addressed all issues raised previously and the changes made have improved the strength of the paper which I think will make a valuable contribution to the literature.